# Positron Emission Computed Tomography Spectrum of Large Vessel Vasculitis in a Tertiary Center: Differences in 18F-fluorodeoxyglucose Uptake between Large Vessel Vasculitis with Predominant Cranial and Extracranial Giant Cell Arteritis Phenotypes

**DOI:** 10.3390/jcm12196164

**Published:** 2023-09-24

**Authors:** Elena Heras-Recuero, Laura Cristina Landaeta-Kancev, Marta Martínez de Bourio-Allona, Arantxa Torres-Rosello, Teresa Blázquez-Sánchez, Iván Ferraz-Amaro, Santos Castañeda, Juan Antonio Martínez-López, Luis Martínez-Dhier, Raquel Largo, Miguel Ángel González-Gay

**Affiliations:** 1Division of Rheumatology, IIS-Fundación Jiménez Díaz, Av. de los Reyes Católicos, 2, 28040 Madrid, Spain; elena.herasr@fjd.es (E.H.-R.); arantxa.torres@quironsalud.es (A.T.-R.); teresa.blazquezs@quironsalud.es (T.B.-S.); jamartinez@quironsalud.es (J.A.M.-L.); rlargo@fjd.es (R.L.); 2Department of Nuclear Medicine, Fundación Jiménez Díaz University Hospital, 28040 Madrid, Spain; laura.landaeta@quironsalud.es (L.C.L.-K.); marta.martinezb@quironsalud.es (M.M.d.B.-A.); lmartinez@quironsalud.es (L.M.-D.); 3Department of Internal Medicine, University of La Laguna (ULL), 38200 Tenerife, Spain; iferrazamaro@hotmail.com; 4Division of Rheumatology, Hospital Universitario de Canarias, 38200 Tenerife, Spain; 5Division of Rheumatology, Hospital Universitario de La Princesa, IIS-Princesa, 28006 Madrid, Spain; scastas@gmail.com; 6Medicine and Psychiatry Department, University of Cantabria, 39008 Santander, Spain

**Keywords:** positron emission computed tomography (PET-CT) with 18F-fluorodeoxyglucose (FDG), large vessel vasculitis, vasculitis, giant cell arteritis, polymyalgia rheumatica

## Abstract

(1) Objective:To assess the spectrum of PET-CT-related large vessel vasculitis (LVV) in a Spanish tertiary center and to determine whether FDG uptake by PET-CT differs between giant cell arteritis (GCA) with predominant cranial or extracranial phenotypes. (2) Methods: The spectrum of patients diagnosed with LVV by PET-CT in a tertiary referral hospital that cares for 450,000 people over a period of two years was reviewed. Moreover, differences in FDG uptake between LVV-GCA with predominantly cranial and extracranial phenotype were analyzed. (3) Results: Eighty patients were diagnosed with LVV by PET-CT. Most were due to systemic vasculitis (*n* = 64; 80%), especially GCA (*n* = 54; 67.5%). Other conditions included the presence of rheumatic diseases (*n* = 4; 3.2%), tumors (*n* = 9; 7.2%) and infections (*n* = 3; 2.4%). LVV-GCA patients with predominant extracranial GCA phenotype were younger (mean ± SD: 68.07 ± 9.91 vs. 75.46 ± 7.64 years; *p* = 0.017) and had a longer delay to the diagnosis (median [interquartile range] 12 [4–18] vs. 4 [3–8]; *p* = 0.006), but had polymyalgia rheumatica symptoms more frequently than those with predominantly cranial GCA phenotype (46.3% vs. 15.4%, *p* = 0.057). When FDG uptake was compared according to the two different disease patterns, no statistically significant differences were observed. However, patients with extracranial LVV-GCA showed a non-significantly higher frequency of vasculitic involvement of lower-extremity arteries. (4) Conclusions: Regardless of the predominant phenotype, LVV identified by PET-CT is more commonly due to GCA in the Spanish population. In these GCA patients, younger age, PMR, and a higher frequency of lower-extremity artery vasculitis suggest the presence of LVV.

## 1. Introduction

Large vessel vasculitis (LVV) in adults covers a wide spectrum of conditions, with giant cell arteritis (GCA) being the most common entity [1]. Although GCA was classically considered to be a disease affecting the cranial arteries [2], imaging techniques have made it possible to identify GCA patients with a predominantly extracranial pattern of the disease. These patients may have few or no cranial ischemic manifestations [3]. They may present with non-specific manifestations such as constitutional syndrome, fever of unknown origin or polymyalgia rheumatica (PMR), which in some cases may have atypical manifestations such as predominantly affecting the lower extremities and pelvic girdle or be refractory to conventional therapy [3,4]. These patients may more frequently present with extracranial ischemic manifestations affecting the upper or lower extremities. In this sense, experts from the European League Against Rheumatism (EULAR) have recommended the use of imaging techniques in patients with suspected LVV [5].

Positron emission tomography-computed tomography (PET-CT) with 18F-fluorodeoxyglucose (18F-FDG) is one of the imaging techniques that has been shown to be useful in identifying patients who present with a predominantly extracranial pattern of GCA [6]. 18F-FDG-PET combined with CT is a functional imaging technique that has demonstrated usefulness for LVV diagnosis due to its ability to detect glucose uptake from the high activity of inflammatory cells in the vessel walls. PET-CT yields a great overview of the extension of vascular inflammation. Moreover, it is useful to exclude other entities such as infection or malignancy [7,8]. Increased FDG uptake is seen after PET imaging in more than 80% of patients with GCA. This is especially true in the case of involvement of the thoracic and abdominal aorta. Additionally, this imaging technique can identify vascular inflammation affecting lower-extremity arteries in patients with GCA [9].

PET-CT is an expensive technique, and this fact constitutes a limitation for its use in typical cases of GCA associated with cranial ischemic manifestation. In these patients presenting with headache or other cranial ischemic manifestations, Doppler ultrasonography and/or biopsy of the temporal arteries are the most commonly used tools for making a diagnosis of GCA. For this reason, in clinical practice, PET-CT is performed when GCA is suspected and cranial ischemic manifestations are not relevant.

Taking into account these considerations, the purpose of the present study was to assess the spectrum of PET-CT-related LVV in a Spanish tertiary center and to determine whether FDG uptake by PET-CT differs between GCA with a predominant cranial or extracranial phenotype.

## 2. Patients and Methods

### 2.1. Study Design and Patient Recruitment

This is a retrospective study conducted at the Fundación Jiménez Díaz University Hospital (Madrid, Spain). Patients who underwent FDG-PET-CT between April 2021 and March 2023 were evaluated. Those with LVV were assessed.

The study procedures acted in accordance with the Helsinki Declaration of 1975, as revised in 2000. Although it was a retrospective study, ethical committee approval was obtained (PIC034-23).

First, we analyzed the spectrum of LVV. In a second step, we evaluated the differences of FDG-PET-CT between GCA patients according to the predominant disease pattern, that is, predominantly cranial or extracranial LVV phenotype.

### 2.2. Study Protocol

#### 2.2.1. Patient Disease Assessment

Fundación Jiménez Díaz University Hospital provides medical care to 450,000 people. During the study period, 1302 patients underwent PET-CT.

The Big Data Department of the Jiménez Díaz Foundation Hospital was asked to carry out an exhaustive search of the patients who underwent PET-CT during the two years of the study. Then PET-CT of patients that included any of the following keywords: “vasculitis”, “large vessels”, “medium vessels”, “vascular wall”, “vascular”, “aortitis”, “vessel inflammation”, “increased FDG vessel uptake”, “polymyalgia rheumatica”, “giant cell arteritis”, “Takayasu arteritis” were reviewed by members of the Divisions of Rheumatology and Nuclear Medicine. Those patients in whom agreement on the presence of LVV was confirmed were included in the study. Subsequently, clinical information was obtained by reviewing medical records. Additionally, rheumatologists evaluated disease diagnosis and demographic and clinical characteristics, as well as laboratory data.

#### 2.2.2. FDG-PET-CT Equipment, Protocol and Interpretation

18F-FDG-PET-CT examinations were performed on an integrated digital PET-CT system (GE Discovery MI3R, with NEMA sensitivity of 7.5 cps/kbq, 3 rings and 15 cm axial field of view). Patients were administered 175–350 mbq (2.5–5.0 MBq/kg) of 18F-FDG after at least a 4h fast. The postinjection rest time was 60min. PET-CT was performed in the supine position, with arms stretched above the head, and scans were acquired from the base of the skull to the femur.

Low-dose CT was performed for PET co-registration (140 kv, 380 ma) followed by PET imaging (1.45 min per bed). Blood glucose levels before tracer injection were <200 mg/mL in all cases.

All PET-CT scans were reviewed by a nuclear medicine physician with experience in LVV. The nuclear medicine physician visually evaluated the characteristics of the distribution of the radiopharmaceutical in four segments of the aorta (ascending, arch, descending thoracic and abdominal) and in five arterial branches (carotids, brachiocephalic trunk, subclavian arteries) and used a visual uptake classification scale. Additionally, information on involvement of the axillary, vertebral, humeral, iliac, and femoral arteries was evaluated. The standardized grading system from 0 to 3 (vascular to hepatic uptake): 0  =  no uptake (≤mediastinum); 1  =  low-grade uptake (<liver); 2  =  intermediate-grade uptake (=liver), 3  =  high-grade uptake (>liver), with grade 2 indicative and grade 3 considered strongly positive for LVV. In all cases, positive PET-CT scans were re-evaluated for confirmation.

### 2.3. Statistical Analysis

Demographic and clinical characteristics of patients with LVV associated with GCA were presented as mean (standard deviation) or percentages for categorical variables. For continuous variables that did not follow a normal distribution, data were reported as median and interquartile range (IQR). Univariable differences between groups were assessed through Student’s *t*-test, the Mann–Whitney U-test, Chi-squared test or Fisher’s exact test according to the normal distribution or the number of subjects. All analyses were conducted using Stata software, version 17/SE (StataCorp, College Station, TX, USA), with a two-sided significance level set at 5%. A *p*-value less than 0.05 was considered statistically significant.

## 3. Results

### 3.1. Spectrum of Patients with LVV

During the period of study, 80 patients were diagnosed with LVV by PET-CT.

Most were due to systemic vasculitis (*n* = 64; 80%), especially GCA (*n* = 54; 67.5%). In this regard, besides five patients diagnosed with Takayasu arteritis, another five patients who showed increased FDG uptake in large vessels were diagnosed with classic Polyarteritis or ANCA-associated vasculitis. Other conditions included the presence of rheumatic diseases (*n* = 4; 3.2%), tumors (*n* = 9; 7.2%) and infections (*n* = 3; 2.4%) (Table 1).

### 3.2. Clinical Differences between GCA Patients According to the Predominant Phenotype

Fifty-four of the 80 patients with LVV were diagnosed with GCA. Since FDG-PET was generally not performed in patients presenting with the classic cranial GCA pattern of the disease, the majority of them, 41 (75.9%) of 54 patients, met the definition of GCA with predominant extracranial phenotype. Most patients with LVV-GCA without cranial ischemic manifestations had not received glucocorticoids before PET-CT examination. However, some of the patients with cranial ischemic manifestations were started on glucocorticoid treatment within the week before PET-CT evaluation.

The differences between patients with a predominantly cranial or extracranial pattern of the disease are shown in Table 2. 

In this sense, patients with predominantly cranial characteristics were older at the time of diagnosis of the disease (mean ± SD: 75.5 ± 7.6 years vs. 68.1 ± 9.9 years, *p* = 0.017). In contrast, patients with extracranial LVV-GCA had a longer delay in diagnosis from symptom onset compared to those with the classic cranial GCA phenotype (median [IQR]: 12 [4,5,6,7,8,9,10,11,12,13,14,15,16,17,18] vs. 4 [3,4,5,6,7,8]; *p* = 0.006). All patients with the predominant cranial phenotype had a diagnosis confirmed by a positive biopsy and/or ultrasound of the temporal artery, while none of the 22 with extracranial LVV-GCA in whom one of these tests was performed had positive results. Despite differences in the presence of cranial ischemic manifestations, as only 1 of 41 patients with predominantly extracranial LVV-GCA complained of headache at diagnosis, patients with extracranial LVV-GCA more commonly had PMR than those with predominant cranial ischemic manifestations (46.3% vs. 15.4%; *p* = 0.057). Similarly, patients with extracranial LVV-GCA more frequently had fever >38 °C at diagnosis, but the difference was not statistically significant. No differences in the presence of constitutional syndrome were observed between patients with predominant cranial or extracranial features. In this regard, it is possible that the high frequency of constitutional characteristics in patients with cranial GCA may have been one of the reasons for requesting PET-CT in these patients. Finally, patients with a predominantly extracranial LVV-GCA pattern had a milder inflammatory response manifested by lower ESR, CRP and platelet levels and higher hemoglobin values.

### 3.3. FDG-PET-CT Differences between Patients with the Classic Cranial LVV-GCA Phenotype and Dose with Extracranial LVV-GCA Phenotype

Patients with a cranial phenotype included in this study also presented vasculitis involvement of extracranial large vessels demonstrated by the presence of a positive finding on PET-CT, and could therefore be considered as having a mixed phenotype. However, in general, in our center patients with predominant cranial ischemic manifestations are diagnosed by ultrasound or biopsy of the temporal arteries. For this reason, PET-CT was generally not performed in these patients. Consequently, most of them were not included in this study. In this sense, in this study, we specifically focused on patients with LVV diagnosed by PET-CT.

Since patients with predominantly cranial and extracranial patterns of the disease showed clinical differences, we aimed to address whether differences in FDG uptake might exist. However, as shown inTable 3,no significant differences were observed. In this sense, subclavian and brachiocephalic involvement was slightly more common in patients with predominant cranial features. In contrast, a non-significant increase in lower extremity artery involvement was found in those with predominant extracranial LVV-GCA.

Regardless of the predominant LVV phenotype, patients with a clinical diagnosis of PMR (*n* = 21) had increased FDG uptake involving predominantly the shoulders, hip and greater trochanter. The frequency of increased of FDG uptake in these sites was significantly more common in patients with a clinical diagnosis of PMR. In this regard, in those without a clinical diagnosis of PMR (*n* = 33), the frequency of increased FDG uptake in these extravascular sites was significantly reduced (shoulders 24.2% vs. 76.2%; hip 18.2% vs. 47.6%; and greater trochanter 9.1% vs. 33.3%, respectively, in those with a clinical diagnosis of PMR ) (in all cases, *p* < 0.05).

## 4. Discussion

This retrospective study that included 80 patients highlights the value of PET-CT in the diagnosis of LVV. This is especially true in patients with GCA without cranial ischemic manifestations who present with nonspecific features or with PMR. Furthermore, the presence of LVV in conditions other than GCA or Takayasu supports the claim that LVV can occur in other systemic vasculitides, rheumatic autoimmune rheumatic diseases, and other unrelated conditions such as tumors or infections. 

PET scanning was reported to have high sensitivity and specificity for the diagnosis of GCA, and at the same time, the negative predictive value of a PET scan for GCA was also very high [10]. With respect to this, the EULAR recommendations for the use of imaging modalities in primary LVV have recently been updated [11]. These recommendations were made by a group of physicians highly experienced in the diagnosis and treatment of LVV. They highlighted the need for early imaging to support the clinical diagnosis of GCA. This should be carried out by experts in imaging diagnosis soon after a diagnosis of GCA is suspected. They considered that a temporal artery biopsy may be an adequate option, in particular when imaging is not readily available or expertise with imaging in GCA is insufficient [11]. However, as previously reported [12,13], the frequency of positive temporal artery biopsy is low in patients with predominantly extracranial pattern of the disease, in particular in those who do not show the classic cranial ischemic manifestation of this vasculitis. EULAR task force experts also pointed out that ultrasound of temporal and axillary arteries should be considered the first imaging technique to be used to assess mural inflammatory changes in patients with suspected GCA [11]. They also said that high-resolution magnetic resonance imaging (MRI) or FDG-PET can be used as alternatives to ultrasound for the assessment of cranial arteries in patients with suspected GCA [11]. Moreover, they emphasized that FDG-PET should be considered the first imaging alternative to ultrasound of extracranial vessels [11]. According to them, FDG-PET, alternatively MRI or CT, can be used to detect mural inflammation or luminal changes of extracranial arteries in patients with suspected GCA [11]. In keeping with these recommendations, FDG-PET-CT was the imaging technique used in our center when a suspicion of extracranial large vessel involvement was suspected. As described [11], this imaging technique also allowed us to identify other conditions different from LVV, such as infections or tumors.

A 5-year systematic literature review (2017–2022) evaluated prospective cohort and cross-sectional studies related to diagnosis, follow-up, outcome prediction, and technical aspects of LVV imaging [14]. The diagnostic accuracy of the data was meta-analyzed in combination with data from a previous systematic literature review. To this end, the authors retrieved 38 studies, giving a total of 81 studies when combined with the 2017 systematic literature review. Taking a clinical diagnosis as the reference standard for GCA, the authors of the study described that FDG-PET for inflammation of cranial and extracranial arteries produced excellent diagnostic performance in GCA with a pooled sensitivity of 76% and pooled specificity of 95% [14].

Increased vascular FDG uptake has been reported in more than two-thirds of GCA patients. As observed in our series, more than 50% of patients with GCA have vasculitis involvement in the thoracic and abdominal aorta and the subclavian arteries [15]. In keeping with our results, vasculitis affecting the femoral arteries has also been reported [15]. PET-CT has also been found to be very useful in making the diagnosis of GCA in patients presenting with atypical manifestations of GCA [16]

PMR may be the initial manifestation of GCA, and imaging techniques, particularly PET-CT, can reveal LVV in at least one-third of patients presenting with isolated PMR [17]. EULAR experts also indicated that FDG-PET and MRI are good tests to identify large vessel involvement in patients presenting with PMR or systemic manifestations only and in whom GCA is a possible diagnosis [11]. This was the case in many of the patients in our series. In this sense, given the availability of imaging techniques in our center, we used PET-CT in these cases. Therefore, a question that needs further investigation is whether PET-CT should be routinely performed in patients with PMR. In this sense, we are in favor of performing a PET-CT in patients with PMR refractory to doses of prednisone of 20–25 mg/day or in patients who present PMR with atypical manifestations such as a predominance of the pelvic girdle manifestations, in patients with PMR who have severe inflammatory low back pain or in those with limb claudication, although arterial pulse asymmetry may not be clinically evident [4]. 

Although PET-CT is not generally used in patients with suspected GCA in whom cranial manifestations are clinically evident, recent studies indicate that PET-CT may also be useful in making the diagnosis of GCA in patients who present with predominant cranial manifestations [18,19]. In this regard, in a prospective study that included 64 patients with newly suspected GCA who underwent PET-CT and temporal artery biopsy within the first 72 h of starting glucocorticoid treatment, the sensitivity of PET-CT was 92% and specificity was 85% using the temporal artery biopsy as a reference. Compared with clinical diagnosis, PET-CT had a sensitivity of 71% and a specificity of 91% [19].

Another question we wanted to address in this study was whether the clinical spectrum of LVV-GCA in our series differed from that in other southern European populations. To do so, we used data published in Reggio-Emilia, Italy, since twenty years ago we reported that the clinical spectrum of biopsy-proven GCA in this Italian region did not differ from that found in a population in Spain [20]. With respect to this, Boiardi et al. evaluated vascular involvement in a series of 127 patients with LVV-GCA diagnosed between 1996 and 2016, who in most cases were followed up at the tertiary center of Reggio Emilia Hospital (Italy) [21]. The differences in terms of LVV involvement between this Italian series and our series are shown in Table 4. The extent of the disease in this series was not established exclusively by PET-CT since the diagnosis of LVV was made by color Doppler ultrasound, CT angiography, magnetic resonance angiography and/or PET-CT scan [21].

In the Italian series, involvement of the carotid, axillary and brachial arteries was significantly more frequent than in our series. In contrast, brachiocephalic involvement was reported less frequently. However, LVV involvement of the thoracic and abdominal aorta and the iliac and femoral arteries was similar (Table 4).

A possible explanation for the differences between these series may be due to the use of different imaging techniques to make the diagnosis of LVV in Reggio Emilia. Furthermore, as previously described, an analysis of the clinical manifestations confirmed that in our series the indication for performing PET-CT in search of GCA was more focused on patients without typical cranial manifestations of GCA. In this sense, we had a longer delay in the diagnosis of GCA than in Reggio-Emilia (10.1 vs. 5.3 weeks). In addition, classic cranial ischemic manifestations of GCA, such as headache (36.2% vs. 24.1%), scalp tenderness (15.7% vs. 3.7%), abnormal temporal artery on physical examination (34.4% vs. 3.7%) or jaw claudication (17.3% vs. 3.7%) was more frequently reported in patients from the Reggio-Emilia series of LVV-GCA.

Consistent with our findings, in a series of 60 GCA patients who underwent PET-CT at the onset of GCA, Malich et al. demonstrated that the most commonly involved arteries were the ascending aorta (72%), followed by the brachiocephalic trunk (62%), aortic arch (60%) and descending aorta (60%) [22].

Although patients with GCA often present with cranial manifestations and extracranial involvement of the large vessels, in some cases, GCA presents with only extracranial involvement of the disease without any classic cranial features of this vasculitis. At this point, we are tempted to establish two well-differentiated phenotypes of the disease. However, we are aware that we must be very cautious in saying that there are two disease phenotypes, as experts may be reluctant to accept this point. However, it has been established that patients with predominantly extracranial GCA, manifested by large vessel involvement of arteries other than the cranial ones, have clinical differences compared to those found in patients with GCA who present the classic cranial manifestations of the disease [3,23]. In this sense, patients with the predominant extracranial LVV-GCA phenotype are generally younger, have a longer duration of symptoms before GCA diagnosis, have more disease relapses, show less commonly a positive temporal artery biopsy and generally require longer duration of treatment. Furthermore, these patients have features of PMR more frequently than those with the predominantly cranial GCA pattern [24]. Therefore, the presence of PMR may constitute a warning sign for the possible presence of an underlying LVV [25,26]. Regrettably, genetic studies so far have not shed light to discriminate between patients with a predominant cranial or extracranial pattern of the disease. In this regard, both biopsy-proven GCA patients with the classic cranial pattern of the disease and those with LVV-GCA without any cranial ischemic manifestations share a strong association with the HLA region, in particular with HLA-DRB1*04:01 [27]. Moreover, the genetic predisposition to develop severe ischemic complications mediated by polymorphisms of the vascular endothelium growth factor observed in biopsy-proven GCA patients with the classic cranial features of the disease was also observed in patients with extracranial LVV-GCA who had ischemic manifestations [28,29]. 

At this point, the experts’ opinion is that a multimodal imaging approach in individuals with LVV that includes the role of ultrasound and MRI is needed to achieve a comprehensive evaluation of vascular disease in patients with LVV. This need forthecombined use of different imaging tests to increase diagnosis accuracy of GCA is supported by a recent review that showed that combined cranial and large vessel ultrasound and PET-CT yielded accuracy for the diagnosis of GCA [30].

On the other hand, correlation of imaging findings with clinical assessment is required [31]. In this regard, the clinical scenario and treatment status of the patient are important at the time of interpreting vascular FDG-PET image findings. A multimodal imaging approach may be useful for the diagnosis and follow-up of patients with LVV. With respect to this, Quin et al. evaluated 84 patients, including 35 with GCA, 30 with Takayasu arteritis, and 19 disease comparators, to establish concordance between the interpretation of magnetic resonance angiography (MRA) and PET for the extent of LVV disease and disease activity, as well as to determine associations between images and clinical evaluation [32]. There was fair agreement between MRA and PET in terms of extent of disease. However, MRA identified a greater degree of vascular involvement than PET due to the detection of both arterial wall and luminal abnormalities. In contrast, in terms of disease activity, inter-rater agreement was higher for PET readings compared to MRA readings, indicating that assessment of disease activity by PET is more reliable than MRA. These findings indicate that MRA and PET capture complementary but different aspects of LVV [32]. Moreover, in another interesting review, Quinn and Grayson confirmed the high sensitivity of PET to identify vascular inflammation. They also highlight that enhanced vascular PET activity may be present in other conditions different from LVV, such as atherosclerosis [33]. However, a possible limitation of PET is that this imaging technique may show evidence of active disease in patients in clinical remission [32,33,34].

In conclusion, regardless of the clinical expression of the disease, PET-CT is an excellent tool to identify the presence of LVV. This technique is especially useful in individuals in whom the classic cranial ischemic manifestations of the disease are not clinically apparent.

## 5. Significance and Innovation

-FDG-PET-CT may be an important diagnostic tool in patients with suspected LVV, in particular of GCA patients without cranial ischemic manifestations of the disease.-No significant differences in the PET-CT large vessel involvement were found between the patients with GCA who were considered to have a predominantly extracranial phenotype when compared with those who also had cranial ischemic manifestations of GCA.

## Figures and Tables

**Table 1 jcm-12-06164-t001:** Spectrum of large vessel involvement in a tertiary hospital of Spain.

Number of patients (%) *n* = 80	
Systemic vasculitis	64 (80)
Primary large vessel vasculitis	59 (73.8)
Giant cell arteritis	54 (67.5)
Cranial phenotype	13 (16.3)
Extracranial phenotype	41 (51.3)
Takayasu arteritis	5 (6.3)
Classic polyarteritis	2 (2.5)
ANCA-associated vasculitis	3 (3.8)
Retroperitoneal fibrosis	1 (1.3)
Rheumatoid arthritis	1 (1.3)
Ankylosing spondylitis	1 (1.3)
Systemic lupus erythematosus	1 (1.3)
Sarcoidosis	1 (1.3)
Infections	3 (3.8)
Tumors	9 (11.3)

**Table 2 jcm-12-06164-t002:** Main clinical features of patients with GCA in whom a PET-CT was performed according to the clinical phenotype: classic cranial LVV-GCA or extracranial LVV-GCA.

	Classic Cranial LVV-GCA Phenotype	Extracranial LVV-GCA Phenotype	*p*
	*n* = 13	*n* = 41	
Age at diagnosis (mean ± SD)	75.5 ± 7.6	68.1 ± 9.9	0.017
Women, n (%)	12 (92.3)	31 (75.6)	0.26
Positive biopsy and/or US of TA, n (%)	13 (100)	0/22 * (0)	<0.001
Delay to diagnosis weeks (median [IQ range])	4 (3–8)	12 (4–18)	0.006
Headache, n (%)	12 (92.3)	1 (2.4)	<0.001
Scalp tenderness (%)	2 (15.4)	0 (0)	0.055
Abnormal temporal artery on physical examination, n (%)	2 (15.4)	0 (0)	0.055
Jaw claudication, n (%)	2 (15.4)	0 (0)	0.055
Polymyalgia rheumatica, n (%)	2 (15.4)	19 (46.3)	0.057
Visual manifestations, n (%)	4 (30.8)	0 (0)	0.002
Permanent visual loss, n (%)	2 (15.4)	0 (0)	0.055
Constitutional syndrome, n (%)	12 (92.3)	32 (78.0)	0.42
Fever > 38 °C	1 (7.7)	11 (26.8)	0.25
Arthralgia/myalgia	6 (46.2)	21 (51.2)	0.75
ESR mm/1st hour (mean ± SD)	101 (71–120)	68 (39–119)	0.12
ESR > 40 mm/1st hour at diagnosis, n (%)	12 (90)	28 (68)	0.15
CRP mg/dL (mean ± SD)	3.8 (2.6–7.9)	3.0 (1.0–7.6)	0.43
Hemoglobin g/dL (mean ± SD)	11.6 ± 1.3	12.3 ± 1.6	0.072
Platelets × 1000/mm^3^ (mean ± SD)	410 ± 134	370 ± 147	0.25

TA: temporal artery. US: ultrasonography. * Number of patients in whom a TA or US of the TA was performed.

**Table 3 jcm-12-06164-t003:** FDG-PET-CT differences between patients with the classic cranial LVV-GCA phenotype and dose with extracranial LVV-GCA phenotype.

	Extracranial LVV-GCA *n* = 41 (%)	Cranial-LVV-GCA *n* = 13 (%)	*p*
Carotid	14 (34.1)	5 (38.5)	0.78
Subclavian	23 (56)	10 (76.9)	0.21
Brachiocephalic	22 (53.7)	9 (69.2)	0.36
Axillary	7 (17)	1 (7.6)	0.66
Humeral	2 (4.9)	0 (0)	0.99
Vertebral	3 (7.3)	2 (15.4)	0.58
Thoracic aorta	31 (75.6)	9 (69.2)	0.45
Ascending	28 (68.3)	9 (69.2)	0.99
Aortic arch	27 (65.9)	9 (69.2)	0.99
Descending	28 (68.3)	9 (69.2)	0.99
Abdominal	20 (48.8)	7 (53.8)	0.75
Iliac	15 (36.6)	3 (23.1)	0.50
Femoral	10 (24.4)	1 (7.6)	0.26

**Table 4 jcm-12-06164-t004:** Comparisons of involved vascular artery at diagnosis between 54 patients with LVV-GCA diagnosed in Madrid (Spain) and 127 patients with LVV-GCA diagnosed in Reggio Emilia (Italy).

	Madrid, Spain *n* = 54 (%)	Reggio Emilia, Italy *n* = 127 (%)	*p*
Carotid	19/54 (35.2)	79/126 * (62.8)	0.001
Subclavian	33/54 (61.1)	86/126 (68.3)	0.35
Brachiocephalic	31/54 (57.4)	28/126 (22.2)	<0.001
Axillary	8/54 (14.8)	46/126 (36.5)	0.004
Humeral	2/54 (3.7)	18/126 (14.3)	0.04
Vertebral	5/54 (9.3)	8/125 (6.4)	0.50
Thoracic aorta	40/54 (74.1)	81/106 (76.4)	0.74
Ascending	37/54 (68.5)	61/107 (57.0)	0.28
Aortic arch	36/54 (66.7)	72/118 (61.0)	0.48
Descending	37/54 (68.5)	73/106 (69.8)	0.96
Abdominal	27/54 (50.0)	68/116 (58.6)	0.29
Iliac	18/54 (33.3)	31/100 (31.0)	0.77
Femoral	11/54 (20.4)	26/100 (26.0)	0.44

Number of positive/number tested *.

## Data Availability

The data presented in this study are available on request from the corresponding author.

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
