# Peer review of "Positron Emission Computed Tomography Spectrum of Large Vessel Vasculitis in a Tertiary Center: Differences in 18F-fluorodeoxyglucose Uptake between Large Vessel Vasculitis with Predominant Cranial and Extracranial Giant Cell Arteritis Phenotypes"

_jcm, 2023, doi:10.3390/jcm12196164_

Round 1
Reviewer 1 Report
This is a retrospective study evaluating FDG-PET in patients with LVV. Despite the large number of the patients included there are a number of issues requiring further clarification.
According to the results 13 patients presented with classical cranial LVV-GCA phenotype and remaining 41 with the extracranial LVV-GCA phenotype. It seems surprising that no patient presented with a mixed cranial and extracranial phenotype which is quite common in GCA
The authors report that 46% in extracranial and 15% in the cranial phenotype suffered from PMR type of symptoms but there is no reference to the findings of FDG-PET in these group regarding the PMT spectrum of disease. How many of these patients had increased tracer intake in synovial structures of shoulder/hip etc, confirming the diagnosis of PMR? And vice versa which was the number of patients without PMR symptoms and FDG-PET picture suggestive of PMR?
Images were evaluated by only 1 clinician which represents a significant restriction of the study as no intraobserver variability is noticed
The recent EULAR recommendations for imaging in LVV indicate ultrasound as a first line method for cranial LVV but it is also suggested that FDG-PET can also be used as a second-line modality for the detection of cranial vessels vasculitis ( Ann Rheum Dis. 2023 Aug 7:ard-2023-224543.RMD Open. 2023 Aug;9(3):e003379). This point should be discussed as considerably strengthens the findings of the current study.
Given the complexity of the field, I also suggest a more critical approach in the discussion with a paragraph commenting on the need of multimodel imaging approach in LVV individuals including the role of ultrasound and magnetic resonance imaging with MRA to achieve the comprehensive evaluation of vascular disease in these patients (Ann Rheum Dis. 2018 Aug;77(8):1165-117, Best Pract Res Clin Rheumatol 2023 Jul 27;101856, Autoimmun Rev. 2023 Jul;22(7):103355, Mediterr J Rheumatol 2021;32(4):363-6, Curr Treatm Opt Rheumatol. 2019 Mar;5(1):20-35)
Reviewer 2 Report
1. line 29: PMR changes into polymyalgia rheumatica.
2. line 53: Positron emission computed tomography changes into Positron emission tomography-computed tomography.
3. line 53 and 102: 18F-FDG=> 18F-FDG.
4. How long did it take for the patient to undergo FDG PET CT and steroid treatment?
Round 2
Reviewer 1 Report
Π